# Super Absorbent Polymers Buried within Dry Soil Attract Formosan Subterranean Termites (Blattodea: Rhinotermitidae)

**Qinxi Xie [1], Zhengya Jin [1], Wei Lin [2], Kena Xue [2], Xuemei Chen [2], Kai Zhao [3], Xiujun Wen [1] and Cai Wang [1,\*]**

[1] Guangdong Key Laboratory for Innovation Development and Utilization of Forest Plant Germplasm, College of Forestry and Landscape Architecture, South China Agricultural University, Guangzhou 510642, China
[2] Foshan Institute of Forestry, Foshan 528222, China
[3] College of Agriculture, South China Agricultural University, Guangzhou 510642, China
\* Correspondence: wangcai@scau.edu.cn; Tel.: +86-20-85280256

**Abstract:** Baiting is one of the main methods to control subterranean termites. Many previous studies showed that subterranean termites avoid making tunnels within dry soil and feeding on dry wood, which may decrease bait infestation and consumption in drought areas. Super absorbent polymers are a group of materials that can retain large amounts of water and improve the moisture content of soil and bait matrices, and therefore may attract termites. In the present study, choice tests were conducted in the laboratory to investigate the aggregation and feeding behaviors of Formosan subterranean termites, *Coptotermes formosanus* Shiraki, in response to the three super absorbent polymers—sodium polyacrylate (Na-PAM), potassium polyacrylate (K-PAM), and poly(acrylamide-co-acrylic acid) potassium salt (P(AM/AA))—that were either placed within soil or filled in the void volume of baiting containers. Under dry-soil (30%-moisture) conditions, termites consumed significantly more wood in the chambers where super absorbent polymers were buried than in the control chambers (super absorbent polymer was not placed within soil). In addition, Na-PAM placed within dry soil significantly increased termite aggregation compared with the control chambers. However, no aggregation or feeding preference was detected when super absorbent polymers were placed within wet soil (60%-moisture). Also, filling super absorbent polymers into the void volume of baiting containers did not attract termites, whether the soil was dry or wet. Our study showed that placing super absorbent polymers within soil around bait stations may increase bait consumption by subterranean termites in drought locations.

**Keywords:** aggregation; bait; *Coptotermes formosanus*; drought; feeding preference; super absorbent polymers

## 1. Introduction

The Formosan subterranean termite, *Coptotermes formosanus* Shiraki, is an economically important pest with a wide distribution in subtropical and temperate areas [1,2]. This pest attacks buildings and wooden structures in urban areas, and caused huge economic losses worldwide [3]. Suszkiw [4] estimated that the annual repair and control cost for *C. formosanus* was >1 billion dollars (USD) in 11 US states. In China, the economic losses caused by *C. formosanus* was ~0.1 billion dollars each year [5]. Unlike many other subterranean termites, *C. formosanus* can also attack live trees. A recent study conducted by Evans et al. [6] showed that large proportions of forests in South Carolina and Louisiana have been infested by *C. formosanus*, which may have resulted in the death of the trees. Since

*C. formosanus* is more likely to attack some tree species (e.g., bitternut hickory, bald cypress, blackgum and sweetgum in Louisiana) than others, it may change the community structure in forests [6].

Baiting is one of the main strategies to control subterranean termites. This method applies a small amount of active ingredients (e.g., hexaflumuron, noviflumuron, etc.) in bait matrices, which can be searched for and consumed by termites, and eventually causes the elimination of the whole colony. Although the success of baiting against termites has been shown over the past 20 years [7], the effectiveness of baits may be reduced under certain environmental conditions. For example, previous studies showed that termites had limited ability to make tunnels in dry soil/sand [8]. Also, low moisture content of wood significantly decreased the feeding activities of termites in the laboratory and field [9,10]. Few studies tried to increase the moisture conditions of bait matrices or soil around bait stations and therefore improve bait attractiveness to termites. For example, Xiong et al. [11] reported that filling the void volume of bait stations with sodium bentonite, a clay material with the ability to absorb and retain large amounts of water, can significantly increase wood moisture and bait consumption by termites under dry conditions. However, filling the bait stations with another water-retaining material, polyacrylamide/attapulgite composite, showed no promise to increase wood consumption by termites [12].

Some patents on termite bait stations claim the use of super absorbent polymers that may improve the moisture of bait matrices [13], but the effectiveness of such potential applications have not yet been investigated. In the present study, we hypothesized that: (1) placing super absorbent polymers within soil around bait stations would attract foraging termites and increase bait consumption; and (2) filling the void volume of bait stations with super absorbent polymers would create a favorable micro-environment for termites and therefore increase their aggregation and feeding activities. To test these hypothesizes, we conducted different choice tests to investigate the preferences of termites in response to super absorbent polymers (either placed within soil or filled in the bait containers) under dry or wet conditions.

## 2. Materials and Methods

### 2.1. Termites

Three colony groups of *C. formosanus* (>500 m apart from each other) were collected in the campus and arboretum of South China Agricultural University (SCAU), Guangzhou, China, using underground monitoring stations containing pine wood sticks [11]. The monitoring stations infested by *C. formosanus* were brought to the laboratory and maintained in 60 L plastic containers for <1 month. Just before the experiment, the wood sticks were gently knocked with each other to extract termites.

### 2.2. Super Absorbent Polymers and Soil (Substrate)

Three super absorbent polymers—sodium polyacrylate (Na-PAM), potassium polyacrylate (K-PAM), and poly(acrylamide-co-acrylic acid) potassium salt (P(AM/AA))—were purchased from online sellers. We chose these particular polymers because they have been widely used to increase the water-holding capacity of agricultural and forest soil [14,15]. The basic information (i.e., manufacturer and water absorption ability) of these super absorbent polymers is shown in Table 1. Top soil (<5 cm) was collected from the arboretum of SCAU and used as the substrate in the present study. A sample of soil was sent to the Laboratory of Forestry and Soil Ecology (College of Forestry and Landscape Architecture, SCAU), and classified as sandy clay loam (54% sand, 23% silt, and 23% clay). The soil was dried at 50 °C for ~2 weeks and sterilized at 80 °C for 3 days. Wooden pestles and mortars were used to ground the dried soil, which was then sifted through a 2-mm sieve to remove the coarse particles. The moisture level of soil was determined using the formula provided by Chen and Shelton [16] as follows: moisture level (%) = [weight of distilled water/(weight of saturated soil − weight of dried soil)] × 100%. The water saturation level of super absorbent polymers was determined using the formula provided by Xie et al. [12] as follows: water saturation (%) = [weight of distilled water added/(weight

of saturated super absorbent polymers − weight of dried super absorbent polymers)] × 100%. One day before the experiments, the required amount of deionized water was added to prepare the super absorbent polymers at the 30% and 60% water saturation, as well as soil (substrate) at the 30% and 60% moisture levels.

**Table 1.** Basic information of each super absorbent polymer (SAP) used in the present study.

| Super Absorbent Polymer | Abbreviation | Water Absorption at Each Saturation Level | | | Brand | Manufacturer |
|---|---|---|---|---|---|---|
| | | 100% [a] | 60% [a] | 30% [a] | | |
| Sodium polyacrylate (cross-linked) | Na-PAM | 795.1 [b] | 477.1 [b] | 238.5 [b] | Sigma | Sigma Aldrich, St Louis, MO, USA |
| Potassium polyacrylate (cross-linked) | K-PAM | 355.2 | 213.1 | 106.6 | Jadreh | Zibo Jadreh Polymer Technology Co, Ltd., Zibo, China |
| Poly(acrylamide-co-acrylic acid) potassium salt (cross-linked) | P(AM/AA) | 183.9 | 110.3 | 55.2 | Sigma | Sigma Aldrich, St Louis, MO, USA |

[a] Water saturation level. [b] Amount of deionized water (g) can be absorbed by 1 g dry powder of super absorbent polymer to reach the 100%, 60%, or 30% water saturation level at room temperature (25 °C).

*2.3. Preference of Termites to Super Absorbent Polymers Placed within Soil*

Two-choice tests were conducted to investigate whether placing super absorbent polymers (Na-PAM, K-PAM, or P(AM/AA)) within dry or wet soil surrounding the baiting containers would increase aggregation and wood consumption of termites. A total of six multiple-choice tests (3 types of super absorbent polymers × 2 moisture conditions) were conducted. Each test was repeated 12 times (4 replicates for each colony group × 3 colony groups).

The bioassay arenas were plastic boxes (153 × 72 × 49 mm [L × W × H]) with three chambers (69 × 50 × 40 mm [L × W × H]). There was a small hole (~5 mm in diameter) at the bottom center of each of the two inner walls of the chambers. A baiting container was prepared by drilling 10 holes (diameter = 5 mm, lined up in 2 rows and staggered distributed) on the wall of a plastic box (diameter of upper side = 41 mm, diameter of bottom side = 31 mm, height = 33 mm). A wood block (pine wood, 20 × 20 × 20 mm) was oven dried (80 °C for 5 days) and weighed using a 0.1 mg electronic balance, and then placed inside the baiting container.

Baiting containers were placed in the side chambers of the bioassay arenas (Figure 1A). Under dry-soil conditions, the baiting container in one side chamber (randomly assigned) was buried using 30% moisture soil, but the top of the baiting container was unburied (Figure 1B). On the other side, soil was first added to partially bury the baiting container in the depth of 1.5 cm (Figure 1C), and 8 g of 30% water-saturated super absorbent polymers (Na-PAM, K-PAM, or P(AM/AA)) was evenly placed on the surface of soil (Figure 1D,E). Additional soil was then added to bury the baiting container (Figure 1F). Similar procedures were conducted to prepare the bioassays under wet-soil conditions, but 60% moisture soil was used as the substrate, and 60% water-saturated super absorbent polymers were placed within soil.

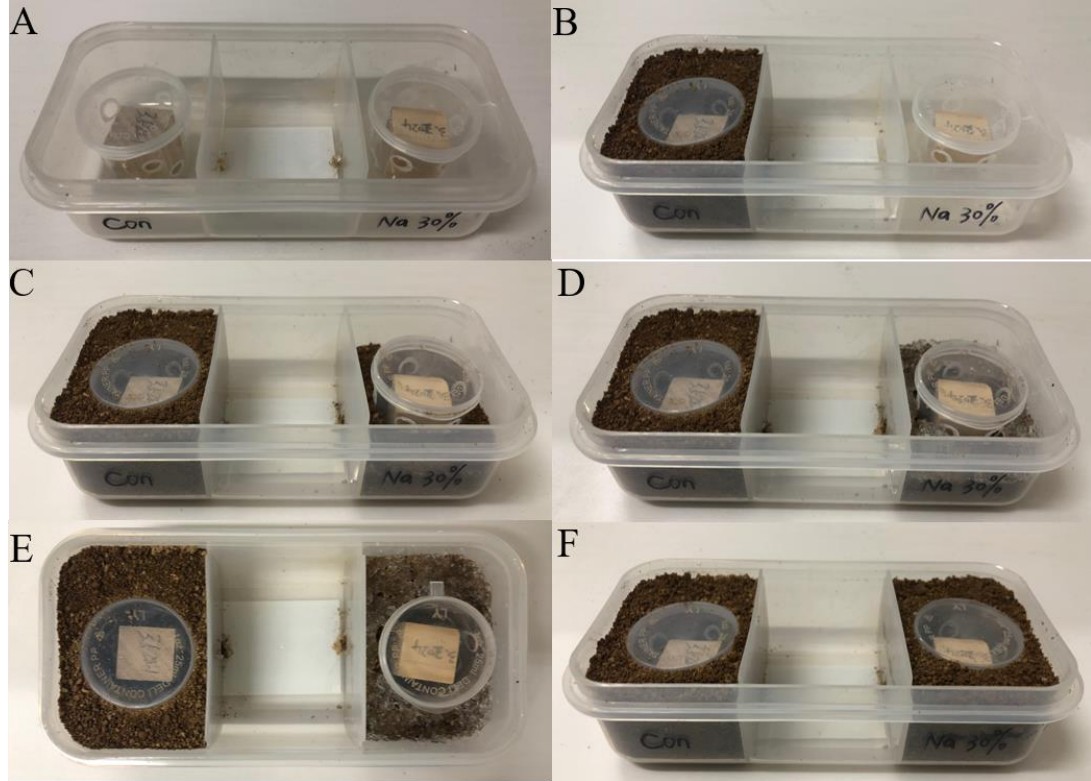

**Figure 1.** Experiment settings to investigate the preference of termites to super absorbent polymers placed within soil. Baiting containers were placed in the side chambers of bioassay arenas (**A**). The baiting container in one side chamber (randomly assigned) was buried using soil, but the top of the baiting container was unburied (**B**). On the other side, soil was first added to partially bury the baiting container in the depth of 1.5 cm (**C**), and 8 g super absorbent polymers were evenly placed on the surface of soil (**D** [side view] and **E** [top-down view]). Additional soil was then added to bury the baiting container (**F**).

One hundred termites (95 workers and 5 soldiers) were placed into the center chamber of each bioassay arena and sealed. The bioassays were placed in the environmental chambers setting at 25 °C under total darkness. On day 21, the bioassay arenas were opened and cotton balls were used to seal the holes on the bottom center of inner walls, and the number of termites in each chamber was counted. The wood blocks in each baiting container were weighed using the 0.1 mg electronic balance as mentioned earlier. These wood blocks were then oven dried (80 °C for five days) and the dry weight was measured. The wood consumption was determined by calculating the dry weight change before and after the experiment. The moisture of wood bock was determined using the formula provided by Xie et al. [12] as follows: wood moisture (%) = [(wet weight after the experiment − dry weight after the experiment)/dry weight after the experiment] × 100%.

### 2.4. Preference of Termites to Super Absorbent Polymers Filled in Baiting Containers

Multiple-choice tests were conducted to investigate the aggregation and wood consumption of termites in response to baiting containers filled with each super absorbent polymers (Na-PAM, K-PAM, or P(AM/AA)), or soil, or unfilled (controls). The tests were conducted either under dry- or wet-soil conditions. In total, there were six multiple-choice tests (3 types of super absorbent polymers × 2 moisture conditions), and each test was repeated 12 times (4 replicates for each colony group × 3 colony groups).

The bioassay arenas were 750 mL plastic boxes (diameter of upper side = 14.1 cm, diameter of bottom side = 11.3 cm, height = 6.5 cm). Baiting containers were prepared as mentioned earlier. Under dry-soil conditions, four baiting containers, which were filled with 30% or 60% water-saturated

super absorbent polymers, or 30% moisture soil, or unfilled, were placed on the bottom of the bioassay arena. Each baiting container was equidistant from the adjacent ones, and their order and cardinal direction were randomly assigned (Figure 2A). Dry soil (30% moisture) was used as the substrate to bury the baiting containers (the tops of the baiting containers were unburied) (Figure 2B). Similar procedures were conducted to prepare the bioassays under wet-soil conditions, but 60% moisture soil was used as the substrate to bury and fill the baiting containers.

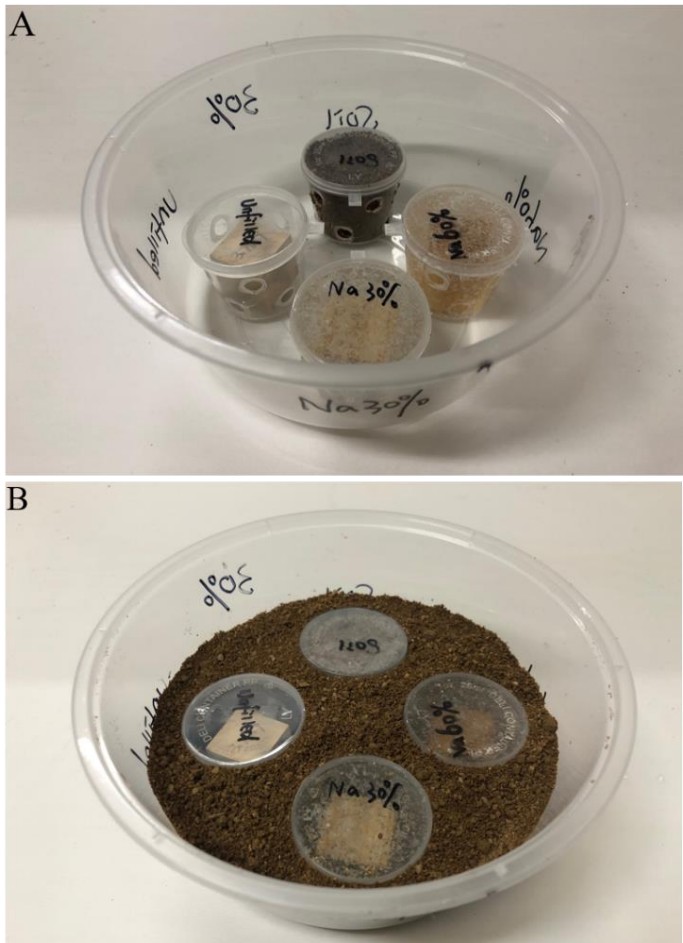

**Figure 2.** Experiment settings to investigate the preference of termites to super absorbent polymers filled in baiting containers. Four baiting containers (either filled with 30% or 60% water-saturated super absorbent polymers, or 30%-moisture soil, or unfilled) were placed on the bottom of the bioassay arena (**A**), and soil was used as the substrate to bury the baiting containers (**B**).

Two hundred and fifty termites (225 workers and 25 soldiers) were released into each bioassay arena and sealed. The bioassays were placed in the environmental chambers setting at 25 °C under total darkness. On day 21, the number of termites aggregated in each baiting container or still within the substrate was counted. The wood consumption and wood moisture were measured, as mentioned earlier.

*2.5. Data Analysis*

For each choice test, the percentage of termites aggregated in each location was calculated. Due to the sum constraint of the percentage data, we conducted the logratio transformation to make the percentages data independent [17]. The transformed data were analyzed using two-way analysis of variance (ANOVA, Proc Mixed, SAS 9.4, SAS Institute, Cary, NC, USA) with colony group as the random factor and location as the fixed effect. Wood consumption and wood moisture were also



compared using two-way ANOVA with colony group as the random factor and chambers or baiting containers as the fixed effect. Tukey's honest significant difference (HSD) tests were used for multiple comparisons after each ANOVA. In all tests, the significance levels were determined at $\alpha = 0.05$.

## 3. Results

### 3.1. Preference of Termites to Super Absorbent Polymers Placed within Soil

The mean survivorship of termites was >75% in all two-choice tests. Under dry-soil conditions, significantly more termites aggregated in the side chambers where Na-PAM was placed compared with the control side (Na-PAM was not placed within the soil) (Figure 3A). Also, wood consumption and wood moisture were significantly higher when Na-PAM was placed within soil (Figure 3B). Similarly, placing K-PAM or P(AM/AA) within soil significantly increased wood consumption and wood moisture compared with the control side, but no effects on termite aggregation was detected (Figures 4 and 5). Under wet-soil conditions, the three super absorbent polymers had no significant effect on termite aggregation and wood consumption (Figures 3–5), though Na-PAM and P(AM/AA) significantly increased the moisture content of wood blocks (Figures 3 and 5).

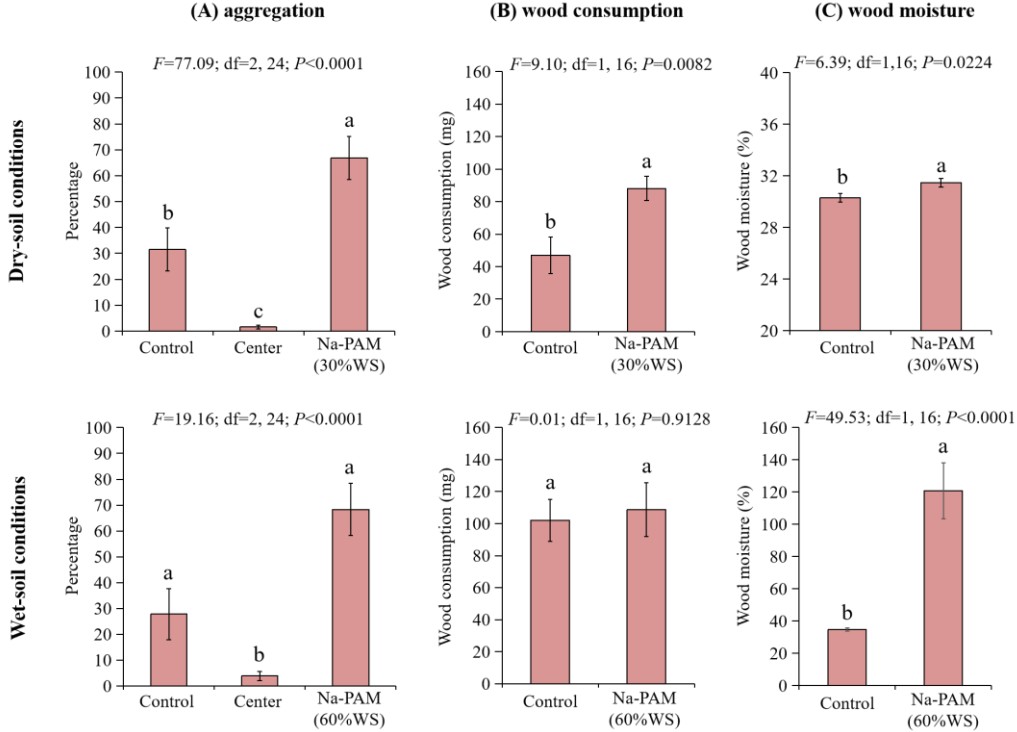

**Figure 3.** Results of the two-choice tests (under dry- or wet-soil conditions) to investigate the preference of termites to sodium polyacrylate (Na-PAM) placed within soil. Percentage of termites in each chamber (**A**), and wood consumption (**B**) and wood moisture (**C**) in the side chambers are presented as mean ± SE. Different letters indicate significant differences ($p < 0.05$). WS represents water saturation level.

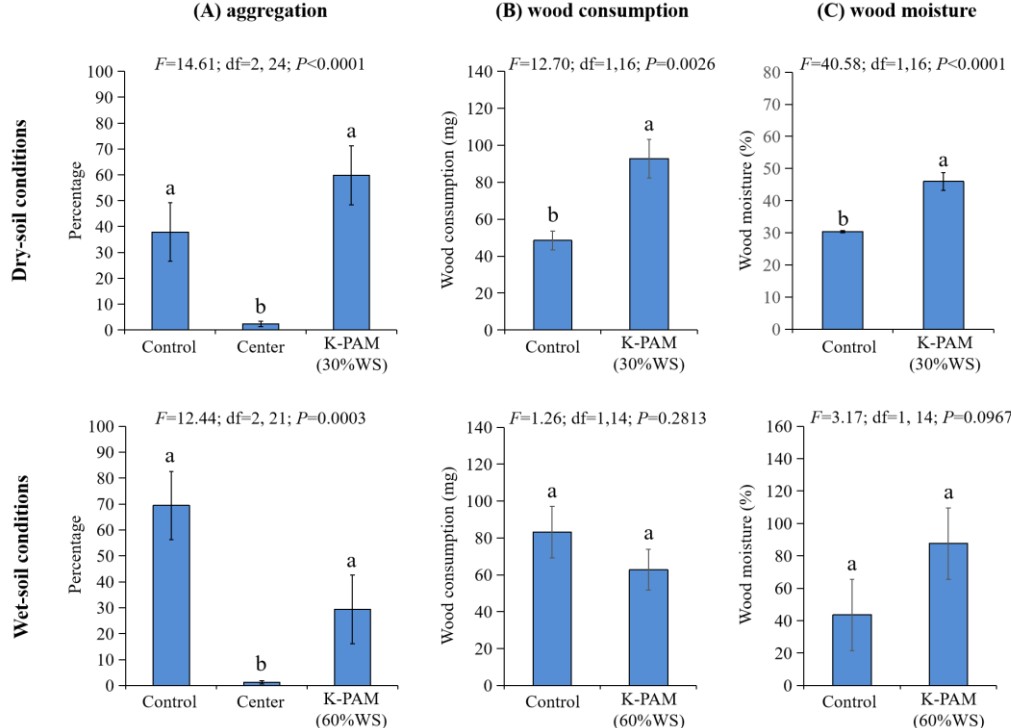

**Figure 4.** Results of the two choice tests (under dry- or wet-soil conditions) to investigate the preference of termites to potassium polyacrylate (K-PAM) placed within soil. Percentage of termites in each chamber (**A**), and wood consumption (**B**) and wood moisture (**C**) in the side chambers are presented as mean ± SE. Different letters indicate significant differences ($p < 0.05$). WS represents water saturation level.

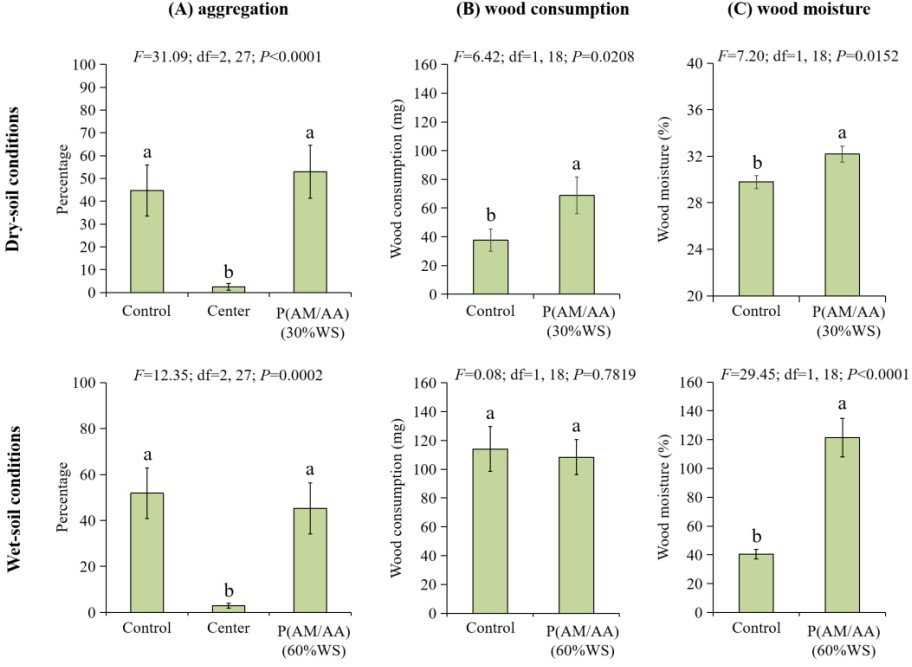

**Figure 5.** Results of the two-choice tests (under dry- or wet-soil conditions) to investigate the preference of termites to poly(acrylamide-co-acrylic acid) potassium salt (P(AM/AA)) placed within soil. Percentage of termites in each chamber (**A**), and wood consumption (**B**) and wood moisture (**C**) in the side chambers are presented as mean ± SE. Different letters indicate significant differences ($p < 0.05$). WS represents water saturation level.

### 3.2. Preference of Termites to Super Absorbent Polymers Filled in Baiting Containers

The mean survivorship of termites was ≥80% in all multiple-choice tests. Under dry-soil conditions, filling the baiting containers with the three super absorbent polymers significantly increased the moisture content of wood blocks (Figures 6–8). However, filling the baiting containers with Na-PAM (30% or 60% water saturation) had no significant effect on the aggregation and wood consumption of termites compared with unfilled or soil-filled ones (Figure 6). Filling the baiting containers with K-PAM significantly decreased wood consumption compared to the unfilled ones, and the least termites were found in the baiting containers filled with 60% water-saturated K-PAM (Figure 7). In addition, 30% water-saturated P(AM/AA) in the baiting containers significantly decreased termite aggregation and wood consumption compared with the soil-filled containers (Figure 8).

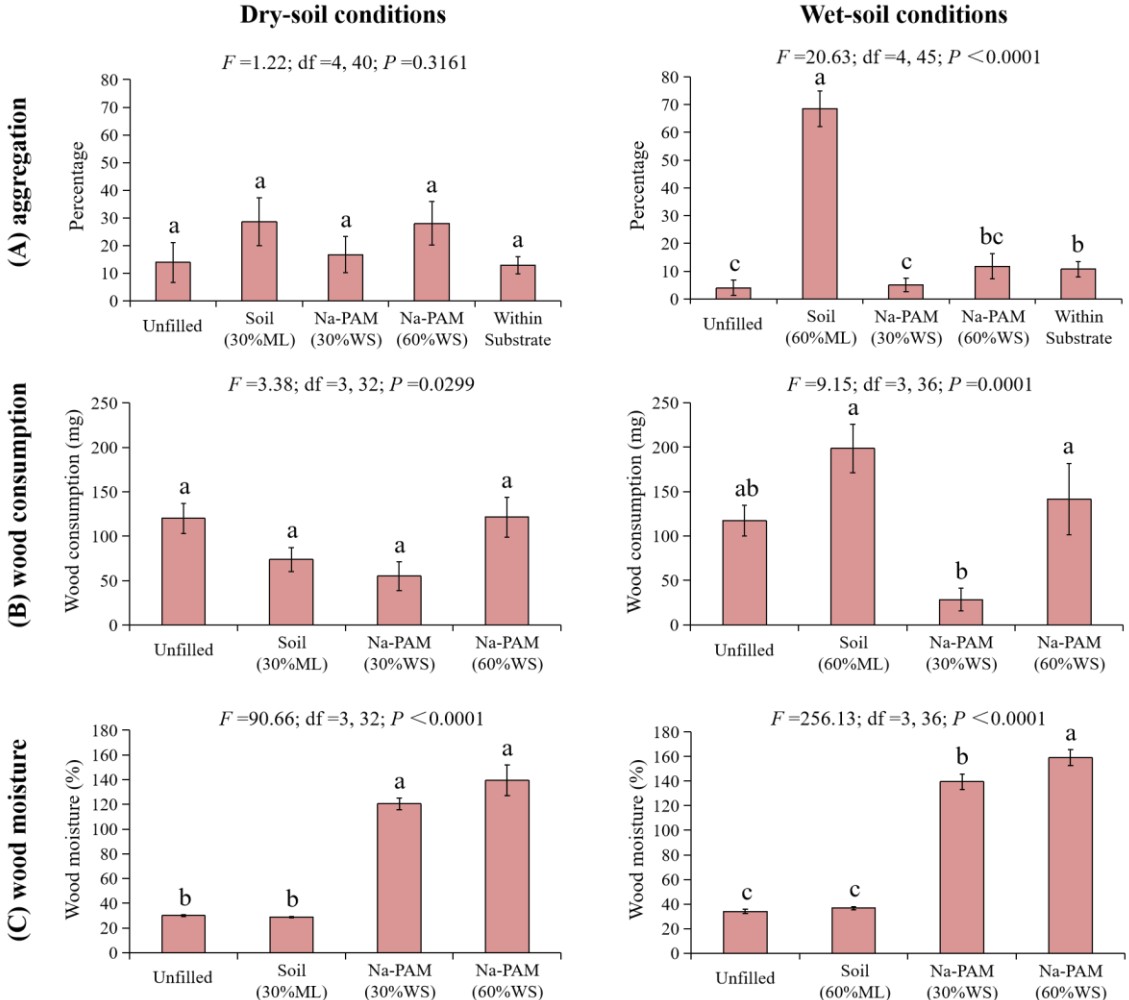

**Figure 6.** Results of the multiple-choice tests (under dry- or wet-soil conditions) to investigate the preference of termites to sodium polyacrylate (Na-PAM) filled in the void volume of baiting containers. Percentage of termites in each location (**A**), and wood consumption (**B**) and wood moisture (C) in each baiting containers are presented as mean ± SE. Different letters indicate significant differences ($p < 0.05$). WS represents water saturation level.

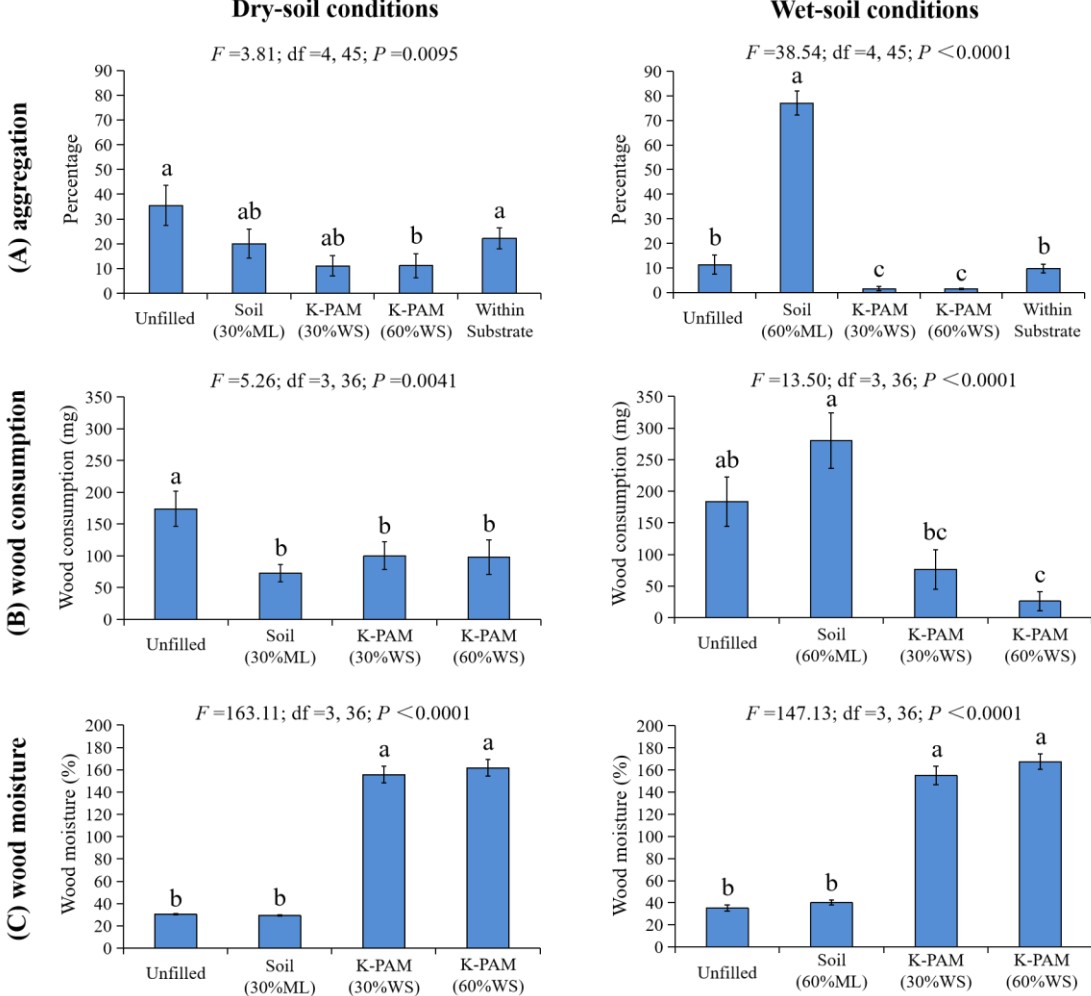

**Figure 7.** Results of the multiple-choice tests (under dry- or wet-soil conditions) to investigate the preference of termites to potassium polyacrylate (K-PAM) filled in the void volume of baiting containers. Percentage of termites in each location (**A**), and wood consumption (**B**) and wood moisture (**C**) in each baiting containers are presented as mean ± SE. Different letters indicate significant differences ($p < 0.05$). WS represents water saturation level.

Under wet-soil conditions, three super absorbent polymers significantly increased the moisture content of the wood blocks (Figures 6–8). However, significantly more termites were found in the baiting containers filled with soil than those that filled with super absorbent polymers (Figures 6–8). In addition, termites consumed significantly more wood in the baiting containers filled with soil than those that filled with 30% water-saturated Na-PAM (Figure 6), and 30% or 60% water-saturated K-PAM and P(AM/AA) (Figures 7 and 8).

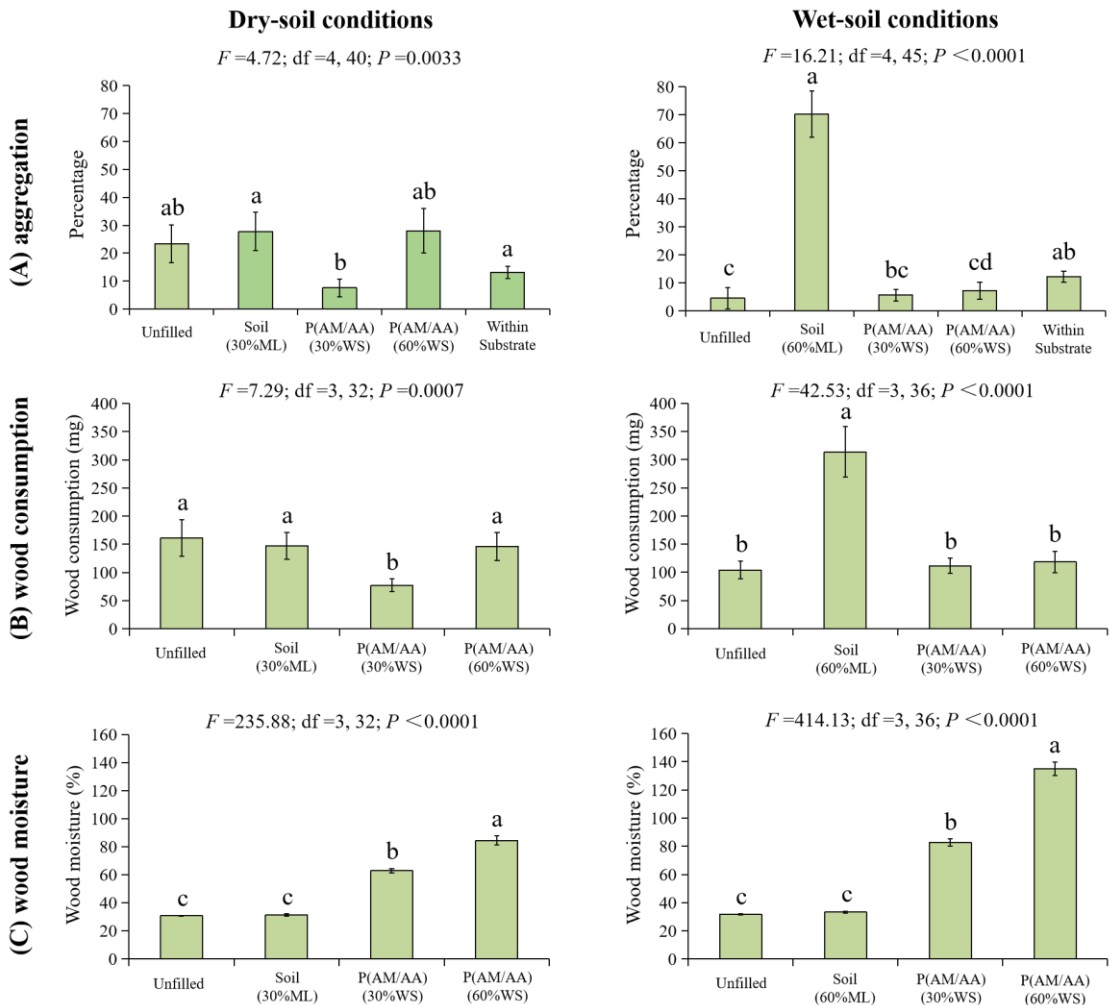

**Figure 8.** Results of the multiple-choice tests (under dry- or wet-soil conditions) to investigate the preference of termites to poly(acrylamide-co-acrylic acid) potassium salt (P(AM/AA)) filled in the void volume of baiting containers. Percentage of termites in each location (**A**), and wood consumption (**B**) and wood moisture (**C**) in each baiting containers are presented as mean ± SE. Different letters indicate significant differences (*p* < 0.05). WS represents water saturation level.

## 4. Discussion

Many previous studies have focused on the termite attractants that may increase bait consumption (as reviewed by Lax et al. [18] and Judd [19]). For example, Chen and Henderson [20] found that *C. formosanus* consumed significantly more filter paper treated with the solution of certain amino acid additives (i.e., D-aspartic acid, L-glutamic acid, and L-aspartic acid) than the control paper. Castillo et al. [21] reported that *Coptotermes gestroi* (Wasmann) consumed significantly more bait matrices composed of cellulose, 3% xylose, 3% casein and cassava powder compared with the control matrices (only containing cellulose), whereas *Coptotermes curvignathus* (Holmgren) consumed more bait matrices (cellulose) when 3% glucose and cassava was added. Suran and Rust [22] reported that 3% xylose in the food (paper discs) significantly increased the intake and horizontal transfer of hexaflumuron among individuals of *Reticulitermes hesperus* Banks, and caused significantly higher termite mortality than that of hexaflumuron alone. Other reported termite attractants include wood-rotting and soil fungi [23–27], decayed wood extract [28], carbon dioxide [29,30], clay materials [11,31,32], and even some sport drinks [33].

Our study showed that super absorbent polymers can act as novel termite attractants. It is worth noting that the three super absorbent polymers tested in our study only increased wood consumption

when: (1) they were placed within soil around baiting containers; and (2) when the substrate (soil) was dry. Evens [34] reported that the subterranean termites *Coptotermes frenchi* Hill made tunnels slowly in dry sand, but they accelerated tunneling by 5 times after a patch of wet sand was discovered. The super absorbent polymers in dry soil may increase the moisture content of adjacent soil so that termites can easily make exploratory tunnels and rapidly locate the food. Also, in our study the presence of super absorbent polymers significantly increased the moisture of wood blocks in baiting containers (Figures 3–5), probably because super absorbent polymers created a micro-environment with high humidity so that wood can absorb more moisture. Termites may also directly transfer water from super absorbent polymers to the wood by evacuating their water sacs, as shown by Gautam and Henderson [35]. Although moisture condition is one of the most significant factors that influence the foraging behaviors of subterranean termites, few efforts have been made to improve baiting systems by increasing the moisture content of soil/bait matrices. Our study showed that placing super absorbent polymers within soil around bait stations may be a simple way to increase bait consumption by termites in dry locations, and it would be valuable to conduct field studies to verify this under natural conditions.

Surprisingly, though the super absorbent polymers' significant increase moisture content of wood blocks in baiting containers, no positive effect on the aggregation and feeding behaviors of termites was found, even when the substrate was dry. A choice test conducted by Gautam and Henderson [36] showed that *C. formosanus* can adapt a wide range of moisture levels (4%–24% moisture) of sand, but significantly fewer termites were found in the chambers containing water-saturated sand (28% moisture). The super absorbent polymers can absorb a large amount of water, and therefore may have created a water-saturated environment inside the baiting containers that inhibited the aggregation and feeding behaviors of termites. Also, we observed that termites cannot easily make tunnels within gel-like super absorbent polymers, which may actually have acted as a barrier for termites to access to the food. Under wet conditions, significantly more termites were found in the baiting containers filled with soil and consumed more wood compared with baiting containers filled with super absorbent polymers or which remained unfilled. In the field, termites usually transport soil or clay into the hollow spaces of trees and bait stations [11,37]. Li and Su [38] claimed that such behaviors help termites to transform the void spaces to "tunnel space during soil displacement". Our results showed that soil filled in void spaces may not only be a byproduct of tunneling excavation and space transformation, but may also enhance the aggregation and feeding activities of termites, and we suggest filling the void volume of bait stations with soil in wet areas to increase bait consumption by termites.

Super absorbent polymers have been widely used in agricultural and forest soil to increase moisture conditions and promote the growth of crops and trees [14,15,39–41]. Our study showed that *C. formosanus* may be attracted by super absorbent polymers placed around plant roots. We suggest paying more attention to termite monitoring and management where super absorbent polymers have been applied, especially under dry-soil conditions.

## 5. Conclusions

This study showed that placing super absorbent polymers within dry soil can significantly increase wood consumption by termites, probably because super absorbent polymers improve the moisture of soil and wood and therefore facilitate the tunneling and feeding behaviors of termites. However, super absorbent polymers placed within wet soil had no detectable effect on termite aggregation and feeding. In addition, filling the void volume of baiting containers with super absorbent polymers cannot attract termites, probably because super absorbent polymers created a microenvironment in the baiting containers that was too wet, inhibiting the foraging behaviors of termites. However, filling the baiting containers with soil significantly increased termite aggregation and consumption under wet conditions. Based on these results, we suggest placing super absorbent polymers within soil surrounding bait stations in drought locations. Also, filling the bait stations with soil may improve bait infestation and consumption in wet locations.

**Author Contributions:** Methodology, Q.X., C.W.; Formal Analysis, Q.X., Z.J., C.W.; Investigation, Q.X., Z.J., K.Z., C.W.; Resources, Q.X., Z.J., X.W., C.W.; Writing-Original Draft Preparation, C.W.; Writing-Review & Editing, Q.X., Z.J., W.L., K.X., X.C., K.Z., X.W., C.W.; Visualization, Q.X., Z.J., W.L., K.X., X.C.; Supervision, X.W., C.W.; Project Administration, W.L., K.X., X.C., C.W.; Funding Acquisition, W.L., K.X., C.W.

**Funding:** This research was funded by National Natural Science Foundation of China (31500530), Science and Technology Program of Foshan (2018AB003861), and Foshan Agricultural Technology Extension Project. The APC was funded by National Natural Science Foundation of China (31500530).

**Acknowledgments:** We sincerely thank Shiping Liang and Hongpeng Xiong (College of Forestry and Landscape Architecture, South China Agricultural University) for valuable helps in termite collection and maintaining.

**Conflicts of Interest:** The authors declare no conflict of interest.

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
