# Peer review of "Super Absorbent Polymers Buried within Dry Soil Attract Formosan Subterranean Termites (Blattodea: Rhinotermitidae)"

_forests, doi:10.3390/f10070591_

Reviewer 1 Report

I read  with interest the manuscript entitled “Super Absorbent Polymers Buried within Dry Soil Attract Formosan Subterranean Termites (Blattodea: Rhinotermitidae)” submitted for publication in Forests journal. 

The authors studied in laboratory the possibility to use the super absorbant polymers introduced in soil located around of bait station, for increase the wood bait consumption by subterrain termites Coptotermes formosanus

In this sense the authors developed two lab. experiments, using on the methodology described suitable in the manuscript. However, I think that it should be more detailed the modality (methodology) used for measure the wood humidity (moisture), before and after the experiments.

The results are clearly and concisely presented. However, please present some information about the termit mortality along the experiments.

 Interpretations of the results in the Discussion are justified and the text is easy to read.

Tables and figures are easy to follow.

 I still have some minor  observations:

Line 41: please insert suplementar the comercial value in USD;

Line 143: please replace ”750 mL” to ”750 ml”;

All Discussion content: please verify and set italic font for all scientific species names. Please setup correctly the rest of the word. 

Author Response

Dear Reviewer, 

 We sincerely thank you for valuable comments on our manuscript. We carefully went through your comments and revised the manuscript accordingly. Following is the response to your comments:

 1. Before the experiments, we placed all wood blocks in 80°C oven drier for 5 days, and the wood blocks are completely dry. As a result, we did not measured wood moisture before the experiment. After the experiment, we measured the wet and dry weight of wood blocks and calculated wood moisture using the method provided by Xie et al. (2019). We provided more details in the revised manuscript:

“The wood blocks in each baiting container were weighed using the 0.1 mg electronic balance as mentioned earlier. These wood blocks were then oven dried (80°C for 5 days) and the dry weight was measured. The wood consumption was determined by calculating the dry weight change before and after the experiment. The moisture of wood bock was determined using the formula provided by Xie et al. [12] as follows: wood moisture (%) = [(wet weight after the experiment– dry weight after the experiment) / dry weight after the experiment] × 100%.”   

 2. We provided termite survivorship information in the results (Line 180 and 205).

 3. We used the commercial value in USD as suggested by the reviewer.

 4. We replaced ”750 mL” to ”750 ml”.

 5. We carefully checked the discussion and set italic font for all scientific species names.

 Thank you!

 Best Regards,

Cai Wang, on behalf of all coauthors

Reviewer 2 Report

This is a well written paper showing the effect of various super absorbant polymers on termite behaviour. I have attached the document with various comments and suggestions for revisions, but they are all very minor.

The only major suggestion is to include why these three particular polymers were chosen. Also there are too many figures and a bit difficult to follow. I suggest perhaps labeling the subfigures a,b,c,etc. and maybe combine the three polymer results into one figure so readers can better see the differences between the effects of the polymers.

Also please check the discussion as there are differences in the formatting throughout. 

Author Response

Dear Reviewer, 

 We sincerely thank you for valuable comments on our manuscript. We carefully went through your comments and revised the manuscript accordingly. Following is the response to your comments:

 1. We chose the three particular polymers because they have been widely used. We mentioned this in the revised manuscript:

 Line 82: Here we chose these particular polymers because they have been widely used to increased the water‐holding capacity of agricultural and forest soil [14, 15].

 2. We labeled the subfigures (a,b,c) in each figure as suggested by the reviewer.

 3. We carefully checked the formatting throughout the manuscript as suggested by the reviewer.

 Thank you!

Best Regards,

Cai Wang, on behalf of all coauthors
